# The Effects of Podophyllotoxin Derivatives on Noncancerous Diseases: A Systematic Review

**DOI:** 10.3390/ijms26030958

**Published:** 2025-01-23

**Authors:** Piotr Strus, Karol Sadowski, Weronika Ploch, Adrianna Jazdzewska, Paulina Oknianska, Oliwia Raniszewska, Izabela Mlynarczuk-Bialy

**Affiliations:** 1Department of Histology and Embryology, Faculty of Medicine, Warsaw Medical University, Chalubinskiego 5, 02-004 Warsaw, Poland; karol.sadowski@wum.edu.pl (K.S.); wera.ploch@gmail.com (W.P.); 2Student Scientific Circle of Rare Diseases at Department of Pediatrics, Hematology and Oncology, Medical University of Gdansk, 80-210 Gdansk, Poland; adrianna.jazdzewska@gumed.edu.pl; 3Student Scientific Circle of Oncology and Radiotherapy at Department of Oncology and Radiotherapy, Medical University of Gdansk, 80-210 Gdansk, Poland; oknianskapaulina@wp.pl; 4Student Scientific Circle of Child and Adolescent Psychiatry, Medical University of Gdansk, 80-210 Gdansk, Poland; oliwiaraniszewska@op.pl

**Keywords:** podophyllotoxin, podophyllotoxin derivatives, noncancerous, genital warts, antimitotic, clinical studies

## Abstract

Podophyllotoxin (PPT) is commonly used for genital warts due to its antimitotic properties and relatively good accessibility since it can be extracted from plants in low-economy countries. However, due to relatively high toxicity, it cannot be used in a systematic way (intravenously). Thus, there is a need to find or create an equally effective derivative of PPT that will be less toxic. Natural PPT is a suitable and promising scaffold for the synthesis of its derivatives. Many of them have been studied in clinical and preclinical models. In this systematic review, we comprehensively assess the medical applications of PPT derivatives, focusing on their advantages and limitations in non-cancerous diseases. Most of the existing research focuses on their applications in cancerous diseases, leaving non-cancerous uses underexplored. To do that, we systematically reviewed the literature using PubMed, Embase, and Cochrane databases from January 2013 to January 2025. In total, 5333 unique references were identified in the initial search, of which 44 were included in the quantitative synthesis. The assessment of the quality of eligible studies was undertaken using the PRISMA criteria. The risk of bias was assessed using a predefined checklist based on PRISMA guidelines. Each study was independently reviewed by two researchers to evaluate bias in study design, reporting, and outcomes. Our analysis highlights the broad therapeutic potential of PPT derivatives, particularly in antiviral applications, including HPV, Dengue, and SARS-CoV-2 infections. Apart from their well-known anti-genital warts activity, these compounds exhibit significant anti-inflammatory, antimitotic, analgesic, and radioprotective properties. For instance, derivatives such as cyclolignan SAU-22.107 show promise in antiviral therapies, while compounds like G-003M demonstrate radioprotective effects by mitigating radiation-induced damage. To build on this, our review highlights that PPT derivatives, apart from anti-genital warts potential, exhibit four key properties—anti-inflammatory, antimitotic, analgesic, and radioprotective—making them promising candidates not only for treating viral infections such as HPV, Dengue, and SARS-CoV-2 but also for expanding their therapeutic potential beyond cancerous diseases. In conclusion, while PPT derivatives hold great potential across various medical domains, their applications in non-cancerous diseases remain limited by the scarcity of dedicated research. Continued exploration of these compounds is essential to unlock their full therapeutic value.

## 1. Introduction

For centuries, observations and research have been conducted on the medicinal properties of natural substances. These organic compounds have a wide variety of medical applications and attract the special attention of scientists due to their good availability and low acquisition cost. Moreover, such compounds, after simple purification, were accessible to people long before modern chemistry allowed large-scale synthesis of any compounds. Their high effectiveness in treating various types of diseases comes from their antiviral, antibacterial, and anticancer properties [1,2,3,4].

While the anticancer properties of PPT and its derivatives have been extensively studied, there is growing interest in their applications for non-cancerous diseases. These include their potential antiviral, analgesic, and radioprotective effects, which have been less explored in comparison to their oncological uses. Investigating these properties is essential, as they may provide new therapeutic options for viral infections, inflammatory conditions, and radiation-induced damage.

*Podophyllum pellatum* is an example of a plant-based substance. Native American tribes used it against helminths as a laxative and remedy for snake bites [5]. Its properties come from the fact that plants from the genus *Podophyllum*, belonging to the *Berberidaceae* family, are a natural source of a chemical substance called podophyllotoxin (PPT). It is a heterocyclic lignan that can be derived from its roots and rhizomes [6,7]. Podophyllotoxin’s structure was first described in the 1930s [8]. The molecule has a five-ring arrangement, with four chiral centers, and contains functional groups: hydroxyl, lactone, acetal, and three methoxy groups, as shown in Figure 1 [8,9].

The specific mechanism of action of PPT is a frequently researched topic that still is only partially elucidated [10,11,12]. Analysis of the molecular structure suggests that the aromatic and dioxol rings can interfere with microtubules [9,13,14,15]. Scientific research proves that it inhibits the polymerization of tubulin and, therefore, stops the production of microtubules. This leads to the impaired formation of the karyokinetic spindle, which is a crucial part of a cell division process, and cell cycle arrest, which gives the PPT antimitotic properties [16]. Moreover, since microtubules are necessary for the movement of intracellular vesicles by kinesin and dynein, interphasal cells are also affected by PPT. We have recently shown that PPT and its derivatives can induce an unfolded protein response and aggresome formation [17].

The 20th century saw a notable shift in the use of podophyllotoxin. The first reported application of PPT in genital warts treatment dates back to 1942 by Kaplan IW [18], and it is being used in this indication, nowadays commonly known as Wartec or Condyline [19,20].

Apart from its use in treating genital warts, PPT has also been explored for managing other dermatological conditions, such as psoriasis, molluscum contagiosum, and keratoacanthoma. These initial studies highlight its potential beyond oncology, particularly in viral and inflammatory skin conditions, although research in these areas remains limited.

With advances in pharmaceutical research, scientists began exploring its potential in cancer and non-cancerous treatments. As an advantage, PPT does not accumulate in the body—its half-life ranges from one to four and a half hours [21]. Like almost every medicine, it also has its drawbacks. Despite promising results from numerous studies demonstrating the high efficacy of PPT against cancerous cells [12,22,23], it turned out to be too toxic for systemic use (e.g., intravenous or intraperitoneally) [24,25]. Preclinical animal studies report that systemic application of PPT leads to its binding to muscle-free tubulin and causes changes in the metabolic pathways of skeletal muscle [26]. Additionally, scientific data is proving its local and systemic toxicity in humans. With its local use, side effects are observed, such as inflammation at the place of application (mostly glans and foreskin), itching, ulceration, pain, burning, dry skin, or bleeding [27,28,29]. PPT should not be used orally, and if accidentally swallowed, dangerous symptoms of poisoning may appear—diarrhea, abnormal heart rhythm, dizziness, bone marrow arrest, and respiratory failure [30,31]. Podophyllotoxin’s toxicity can manifest itself even after topical application in high doses, resulting in coma with major neurologic, hematologic, and hepatic complications [32]. There are indications that podophyllotoxin can be used for pregnant women, but it may also affect fertility and be teratogenic [33]. Thus, Condyline is approved with the description: Do not use during pregnancy or breastfeeding [20].

Despite significant progress in understanding the therapeutic potential of podophyllotoxin and its derivatives, their applications in non-cancerous diseases remain underexplored. Existing reviews primarily focus on anticancer applications, leaving a knowledge gap in evaluating their broader pharmacological potential, particularly in the context of antiviral, analgesic, and radioprotective effects. Therefore, despite Podophyllotoxin being effective towards cancer cells [34], its antineoplastic use is limited to condyloma acuminata topical treatment [28]. This limitation arises from its selective cytotoxic effects on rapidly proliferating cells, such as those found in warts caused by the human papillomavirus (HPV), which parallels its potential antineoplastic mechanisms as discussed later in this manuscript. Other skin conditions, in which PPT use was introduced, include psoriasis [35], molluscum contagiosum [36], and keratoacanthoma [37].

To find better alternatives for PPT, scientists focused on the synthesis of its derivatives. Structural modifications of PPT led to the discovery of compounds such as teniposide and etoposide [38]. Contrary to the precursor compound, their antimitotic mechanism of action consists of the interaction with an enzyme—topoisomerase II, whose inhibition causes breaks in DNA [39].

Etoposide is a semi-synthetic derivative, a D-glucose PPT’s glycoside, as shown in Figure 2. It is commonly used to treat retinoblastoma, ovarian cancer, testicular cancer, acute leukemia, malignant granuloma, and small-cell lung cancer [40,41,42,43]. Moreover, it became the drug of first choice for the treatment of many malignancies such as testicular and small cell lung cancer [44]. Etoposide’s systemic application was shown to induce a complete response in up to 25% of previously treated patients with acute nonlymphocytic leukemia [45].

Another PPT derivative, teniposide, as shown in Figure 2, is chemically similar to etoposide. It is distinguished only by the thienyl group, whereas etoposide has a methyl group. It is mainly used to treat Hodgkin’s lymphoma, acute lymphocytic leukemia, bladder cancer, and immature neuroblastoma [46].

Etoposide and teniposide are both derivatives of podophyllotoxin. The difference occurs in unmethylation along with an additional group of D-glucose derivatives (with methyl in the case of etoposide or thienyl in teniposide)—the differences are presented in Figure 1 and Figure 2.

Even though PPT and its derivatives are widely researched as anticancer agents, we studied whether the derivatives are also used in non-cancerous diseases, choosing mainly clinical studies with such compounds.

We aimed to investigate the effects of PPT and its derivatives in the case of non-cancerous diseases [47].

Therefore, using database searches using specific keywords, we examined scientific data on this previously unexplored topic, formulated conclusions, and conducted an analysis regarding future applications of PPT and its derivatives in non-cancerous diseases. Only original research papers published within the last 10 years were qualified for further analysis.

This paper aims to systematically review and analyze the latest studies on PPT and its derivatives, focusing on their applications in non-cancerous diseases. By highlighting their therapeutic advantages, limitations, and potential, we seek to underscore the need for further research in this underexplored area to fully understand and expand their clinical utility.

## 2. Methods

This systematic review was conducted following the guidelines outlined by the PRISMA (Preferred Reporting Items for Systematic Reviews and Meta-Analyses) statement [48]. On 15 January 2025, we searched the MEDLINE/PubMed, Embase, and Web of Science databases for papers about podophyllotoxin and its derivatives and their medical applications. Additionally, we consulted the reference lists of included studies to identify potentially relevant papers. No organizational registers or other external databases were used. We selected papers that included “podophyllotoxin” or “PPT” and “non-cancerous” or “disease” or “antiviral” in their title or abstract. The combined search strategy is presented in Table 1.

We narrowed our search and added time restrictions to publications from January 2013 to January 2025. Using this algorithm, we identified 533 records after removing duplicates. All of the 533 papers eligible for abstract-based screening were randomly divided into two subgroups: Subgroup A, consisting of 266 records and Subgroup B, consisting of 267 records. We used EndNote Web to collect the data. Data collection from reports was conducted independently by two reviewers. No specific tools for data confirmation or investigator correspondence were utilized. Each subgroup was analyzed by a different pair of researchers. Two independent researchers reviewed every record and decided if it fulfilled all the inclusion criteria listed in Table 2. Based on this assessment, each of them made their own decision if the article should be included in our study. If they agreed, the article was included in this review. In case of any dispute, the third independent researcher made the final inclusion decision. The inter-rater agreement was assessed using Cohen’s kappa coefficient. The reviewers evaluated the articles based on ‘yes’ or ‘no’ criteria. The calculated kappa values were 0.801 for Subgroup A and 0.816 for Subgroup B, indicating excellent agreement in both subgroups. No automation tools were used in the selection process; all records were screened manually. The exact exclusion criteria are listed in Table 2. We did not count how many articles were excluded by each of the criteria because numerous studies fulfilled two or more of them. Only original research papers were qualified for further analysis. Studies were grouped for synthesis based on their primary focus on the non-cancerous applications of podophyllotoxin and its derivatives, including antiviral, anti-inflammatory, analgesic, and radioprotective effects. Data items included outcomes related to antiviral, anti-inflammatory, analgesic, and radioprotective effects of PPT derivatives. Participant characteristics, intervention details, and funding sources were also collected when reported. We rejected records only about the anticancer activity of PPT and its derivatives, as well as articles without available abstracts or with abstracts written in a language other than English.

We created a table of evidence and assessed 63 full-text articles for eligibility. Furthermore, we excluded records without full text available in English, and unfinished articles. We only included adequate-quality articles with full text available in English. As an outcome of the selection process, there were 44 articles qualified for the final synthesis. No specific examples of excluded studies are highlighted, as all exclusions were based on predefined eligibility criteria, such as language or focus on anticancer activity enlisted in Table 2. Selected articles were classified into folders based on the specific subtopics described. This systematic review focuses on identifying and categorizing the applications of podophyllotoxin and its derivatives in non-cancerous diseases. Given the heterogeneity of the included studies, effect measures such as mean differences or risk ratios were not uniformly applicable. Instead, the findings were synthesized narratively to provide a comprehensive overview of the therapeutic potential and limitations of these compounds. A tabular presentation of findings was used to summarize the results and highlight key trends. Due to heterogeneity in study designs, outcomes, and measures reported, narrative synthesis was employed to integrate findings. Subgroup analyses and sensitivity analyses were deemed inappropriate due to data limitations. Reporting bias was not systematically assessed, as the data were insufficient for funnel plot generation. Similarly, the certainty of evidence was not formally graded, given the exploratory nature of this review and the absence of standardized outcome measures across studies. The Preferred Reporting Items for Systematic Reviews and Meta-Analyses (PRISMA) 2020 flow diagram was used to illustrate the extraction of studies through the different phases of this review, Figure 3 [48].

## 3. Results and Discussion

### 3.1. Results

#### 3.1.1. Toxic Effects of PPT and Its Derivatives

##### Toxicity of PPT to Non-Cancerous Cells

The first indication for the use of podophyllotoxin were genital warts, which are formed from keratinocytes (human skin cells). Studies in cellular models in the 1970s and 1980s focused on the characterization of the mechanisms of action, primarily in the human keratinocyte cell line HaCaT.

Podophyllotoxin (PPT) exhibits various mechanisms of inducing cell toxicity such as metabolic stress by mitochondrial oedema, and disruptions in protein metabolism in the endoplasmic reticulum and Golgi apparatus. Moreover, meiosis inhibition caused by destruction of chromosome alignment, spindle deformation, and DNA damage, especially in oocytes, were noticed. PPT is associated with epithelial cell proliferation and an increase in TNF-α, which causes general inflammation in the tissues. Subsequently, inflammation increases the overall expression of cytochrome c, caspase-8, caspase-9, and caspase-3, which contribute to caspase-induced apoptosis [10,49].

Similar conclusions were drawn when investigating the effects of feeding rats with PPT obtained from Juniperus sabina. Despite the noticeable effect of caspase activation, the TNF-α increase was correlated with the administered dose of podophyllotoxin, which also induced apoptosis. Adverse effects were observed when considering the rats’ fertility, as sperm mobility was inhibited [50].

PPT has been investigated for its effects on animal oocyte maturation in multiple studies [10,51,52]. The exposure to PPT was significantly associated with inhibition of meiosis by spindle deformations. Blocked meiosis induces oxidative stress in the mechanism of DNA damage and severe chromosome arrangement disorders. Moreover, Annexin-V, which binds to phosphatidylserine as a signal for phagocytes, shows increased levels in podophyllotoxin-treated groups, indicating the early stages of apoptosis. Exposure to PPT is strongly correlated with the abnormal distribution rate of the Golgi apparatus and Endoplasmic reticulum [10,51,52].

Another major toxicity event, especially in the liver, was noticed when a study on Wistar rats was performed. Podophyllotoxin-4-O-D-glucoside, Podorhizol, PPT, Podophyllotoxin, and 3′,4′-O-O-Didemethylopodophyllotoxin were identified as the compounds essentially responsible for hepatotoxicity after using them on Wistar rats. This was manifested by particular changes in hepatic tissue: hepatocytes’ edema, cytoplasmic vacuolization, and increased intrahepatic pigmentation. Aside from hepatotoxicity, other symptoms in rats, such as gastrointestinal irradiation with diarrhea were present. 3′,4′-O-O-Didemethylopodophyllotoxin was found to increase energy consumption, which is associated with a higher hepatotoxicity occurrence, which was behaviorally seen as different resting degrees and dyspnoea in rats [53].

More findings suggest that PPT mediates hepatotoxicity via the C5a/C5aR/ROS/NLRP3 axis and inhibition of autophagy through the cGMP/PKG/mTOR pathway. These mechanisms collectively lead to pyroptosis, a form of programmed cell death. Evidence from studies in rats indicated that PPT administration increased serum markers such as AST/ALT and oxidative stress indicators while reducing antioxidant levels (e.g., SOD and T-AOC). Histopathological examination revealed liver damage characterized by congestion and inflammatory infiltration. Proteomic and phosphoproteomic analysis further highlighted a significant upregulation of inflammation-related proteins (e.g., NLRP3, caspase-1) and an inhibition of autophagy-related markers, suggesting a direct link between PPT toxicity and these pathways. These findings elucidate key molecular mechanisms underlying PPT-induced liver damage, providing critical insights for the safer use of PPT in clinical settings [54].

##### Comparative Analysis of PPT Derivatives: KL3

Our team has been studying PPT and its derivatives for years. We have synthesized PPT derivatives, one of which is KL3. As part of this study, KL3 and PPT’s effect on non-cancerous cells were investigated. Potential PPT derivatives are expected to increase the expression of caspases to induce apoptosis.

Research comparing the podophyllotoxin derivative KL3 to PPT was conducted on the Human Keratinocyte cell line (HaCaT) to determine its effects. KL3, at concentrations greater than 10 µM, was found to be a better agent (less toxic and enabling recovery from metabolic stress with time) than PPT in this case by activating caspase-9 [17]. Nevertheless, KL3 did not cause necrosis of the cells in contrast to PPT [17].

##### Impact on Organelles

Previous studies examined the effect of PPT on individual organelles [51]. Particular attention was paid to the study of the organelles of mouse oocytes during meiosis. PPT, dissolved in DMSO, was administered at various concentrations to discern its effects comprehensively. The study found a notable reduction in the developmental competence of mouse oocytes post-PPT exposure [10]. Distinct alterations in organelle distribution were observed, reflecting compromised cellular functions.

Specifically, the endoplasmic reticulum (ER) failed to accumulate around the spindle periphery after PPT exposure, indicating a disruption in protein synthesis during oocyte meiotic maturation. This anomaly suggests potential interference with critical cellular processes, likely influencing subsequent developmental stages. Furthermore, abnormal distribution patterns of the Golgi apparatus were noted, accompanied by diminished localization of Rab11a-related vesicles, signifying compromised vesicle-based protein transport—a fundamental mechanism governing cellular homeostasis [51].

Moreover, aberrant lysosome accumulation within the cytoplasm of PPT-exposed oocytes was identified, suggesting disrupted protein degradation pathways [55]. This disruption could lead to intracellular accumulation of harmful substances, potentially exacerbating cellular stress responses. Additionally, perturbations in mitochondrial distribution were evident, indicative of mitochondrial dysfunction after PPT exposure. Mitochondrial dysfunction is particularly concerning, given its pivotal role in ATP production and cellular energy homeostasis.

PPT exposure induces significant disruptions in organelle dynamics during mouse oocyte meiotic maturation. These findings highlight the intricate interplay between PPT and cellular processes, emphasizing the necessity for further clarification of the underlying molecular mechanisms. Understanding the precise mechanisms through which PPT influences organelle dynamics is crucial for evaluating its safety profile and exploring potential therapeutic applications in clinical contexts [10].

##### Mitotic Spindle Inhibition

PPT and its derivatives have emerged as pivotal players in cellular dynamics, particularly in their remarkable ability to modulate the spindle apparatus during cell division. The spindle, a microtubule-based structure, orchestrates the precise segregation of chromosomes, a process critical for cellular proliferation. Herein, we delve into these compounds’ consequential cause-and-effect relationship of spindle inhibition, unraveling their profound impact on cellular events.

PPT, renowned for its antimitotic properties, disrupts the spindle’s functionality, leading to the arrest of cell division [56]. This interference stems from the compound’s potent interaction with tubulin, specifically targeting the colchicine domain located between α- and β-tubulin [9,57], precisely at the same site as colchicine. The colchicine site is mostly buried in the intermediate domain of the b subunit [9,58]. Tubulin is a key protein constituent of spindle microtubules. The binding alters the microtubule dynamics, inhibiting their polymerization and ultimately impeding spindle formation.

Remarkably, modifications of PPT, such as epimerization in specific positions, exhibit a nuanced influence on spindle inhibition. These alterations induce changes in the compound’s spatial orientation, affecting its interaction with tubulin and subsequently modulating spindle dynamics. The consequential reduction in spindle activity is associated with a noteworthy decrease in cellular proliferation.

Emerging evidence suggests that PPT has potential therapeutic applications beyond its well-documented role in mitotic spindle inhibition, particularly as an anti-leishmaniasis agent. Leishmaniasis, a parasitic zoonotic disease, could benefit from PPT’s ability to disrupt microtubule formation, presenting a novel mechanism of action when used alongside traditional anti-inflammatory topical treatments. This prospect is supported by promising findings from Ghayour et al., who demonstrated that PPT, at a concentration of 200 µg/mL, exhibited 83% lethality against Leishmania major promastigotes 48 h post-cultivation. In comparison, podophyllin achieved only 58% lethality under similar conditions, highlighting PPT’s superior efficacy. Additionally, PPT has drawn significant attention for its antiparasitic activities against other pathogens, including Trypanosoma brucei and Giardia. Further investigations into PPT’s toxicological and pharmacological profiles are warranted to advance its clinical utility in parasitic disease treatment [59].

In summary, PPT and its derivatives intricately intervene in the spindle apparatus, revealing a captivating tale of molecular interactions and their cascading effects on cellular processes [60]. Understanding the detailed mechanisms of spindle inhibition by these compounds not only enriches our knowledge of cell biology but also offers promising avenues for developing targeted antiproliferative therapies.

##### Embryology

PPT and its derivatives, recognized for their medicinal properties, notably impact embryonic development. Investigations into its effects on embryonic development reveal concerning outcomes, including disruptions in spindle organization, chromosome alignment, and DNA integrity during early embryo development [10]. In a study examining fertilized oocytes using a mouse model, exposure to PPT resulted in adverse effects on zygote cleavage, characterized by impaired spindle organization and chromosome alignment during the metaphase of the first cleavage. PPT exposure induced oxidative stress, as evidenced by elevated reactive oxygen species (ROS) levels. Remarkably, PPT-exposed embryos exhibited increased γH2A.X and Annexin-V signals, indicating embryonic DNA damage and early apoptosis [10]. These findings underscore the detrimental impact of PPT on spindle formation, chromosome alignment, and DNA integrity during early embryonic development.

#### 3.1.2. Therapeutic Effects of PPT and Its Derivatives

##### Antiviral Properties

Podophyllotoxin derivatives are commonly used as anticancer chemotherapeutics but also in treating genital warts caused by Human papillomavirus (HPV). HPV infections are usually manifested in genital and plantar warts, which are transmitted by a sexual route [61]. Those appear when the virus infiltrates the epithelium in the basal and spinous layers, causing significant bulges in the form of warts. HPV types 1–4 replicate in epithelium cells, causing plantar warts. However, the other 6, 11 types are responsible for generating mucosal warts [62]. Most of the warts can disappear within the three years, but due to aesthetics and pain-free comfort, treatment is usually applied and recommended to prevent further spread. This includes mainly cryotherapy, electric coagulation, surgical removal, laser therapy, and podophyllotoxin derivative ointments [63].

Podophyllotoxin is used in cases of HPV infections due to its antiproliferative and proapoptotic properties. PPT might be used alone or in combination with other derivatives such as imiquimod. Both of these drugs are equally efficient [64,65]. Combination therapy using cryotherapy along with podophyllotoxin derivative ointments, such as imiquimod, is a standard practice of care for genital warts [66,67,68].

Nevertheless, some data imply that PPT might be more effective in clearing keratinized anogenital warts [69]. However, in some cases, PPT does not give positive results, therefore opting for imiquimod with cryotherapy might be considered a better option [66].

Moreover, Cantharidin–Podophyllotoxin–Salicylic acid (CPS) therapy was more effective than liquid nitrogen monotherapy and was associated with lower pain levels than Long-Pulsed Nd:YAG Laser [68,69,70]. High-intensity 20 MHz ultrasound following Imiquimod 5% cream was also reported as a good treatment option [71]. The abovementioned therapeutic methods are not always as effective as expected, therefore alternative options have to be considered. Tomić et al. presented a case of meatal intraurethral warts with unsuccessful treatment by cryotherapy, laser therapy, imiquimod, electrosurgical resection, and trichloroacetic acid, which was subsequently managed by 5-fluorouracil cream [72].

Podophyllotoxin and its derivatives are commonly used in HPV infections due to their antiproliferative and proapoptotic properties.

Recently, promising results have been reported for a new derivative, cyclolignan SAU-22.107, which decreased the viral yield of the dengue virus by altering the DENV genome replication or translation [73]. Additionally, an experimental study showed that podophyllotoxin exhibits neuroprotective activity in central nervous system (CNS) viral infections. It interacts with amino acid residues Gln189, Met49, Phe140, Glu166, and Asn142, allowing it to bind to the SARS-CoV-2 main protease. Podophyllotoxin also binds to the Cys480 residue of the SARS-CoV-2 spike’s receptor-binding domain. These interactions may inhibit viral replication by modulating the virus’s ability to bind to host cells, thus providing a potential therapeutic avenue for treating SARS-CoV-2 infections [74].

##### Analgesics and Anti-Inflammatory Properties

PPT and its derivatives, known for their diverse pharmacological properties, have recently gained attention for their intriguing anti-inflammatory and analgesic properties. The anti-inflammatory effects of these compounds are closely linked to their modulation of key molecular pathways involved in immune responses [75]. At the molecular level, PPT has been shown to inhibit pro-inflammatory mediators, such as cytokines and chemokines, thereby attenuating the inflammatory cascade. This inhibition correlates with a reduction in the activation of transcription factors like NF-κB [76], which are pivotal in orchestrating inflammatory responses. The consequential suppression of these pathways contributes to the observed anti-inflammatory efficacy of PPT and its derivatives.

Beyond their anti-inflammatory prowess, these compounds also exhibit noteworthy analgesic effects. The attenuation of inflammatory mediators intersects with the modulation of pain pathways, reducing pain perception. This dual action on both inflammation and pain highlights the potential of PPT and its derivatives in developing novel analgesic agents [75,77].

In summary, the anti-inflammatory and analgesic properties of PPT and its derivatives stem from their intricate molecular interactions, particularly in the regulation of key inflammatory pathways. These findings not only deepen our understanding of the pharmacological profile of these compounds but also offer promising avenues for their utilization in the development of therapeutics targeting inflammation and pain [75].

##### Radiation Protection

Radiation exposure, whether intentional or accidental, presents substantial risks to human health, including pulmonary inflammation, fibrosis, and hematopoietic suppression [78]. Developing effective radioprotective agents is crucial for the mitigation of these adverse effects. PPT and its derivatives, particularly formulations such as G-003M [77] and G-002M, have demonstrated promising radioprotective capabilities in preclinical investigations [55]. The radioprotective properties of PPT and its derivatives stem from their ability to scavenge reactive oxygen species (ROS) generated by ionizing radiation [79]. These compounds possess potent antioxidant activity, which helps mitigate oxidative stress-induced damage to cellular structures and DNA [77]. Additionally, PPT derivatives have been found to modulate various signaling pathways involved in the cellular response to radiation, including those regulating inflammation, apoptosis, and DNA repair processes. Furthermore, studies have elucidated the ability of PPT derivatives to enhance the activity of endogenous antioxidant enzymes, such as superoxide dismutase and catalase, thereby bolstering the cellular defense mechanisms against radiation-induced oxidative injury [77].

G-003M, a combination of PPT and rutin, exhibits notable radioprotective effects against gamma radiation-induced pulmonary damage [77]. Rutin, a flavonoid belonging to the flavonol group found in plants, demonstrates the ability to absorb UV radiation. It acts protectively on the skin by reducing DNA damage and the level of free radicals. Rutin’s strong antioxidant properties [80] further enhance its potential as a protective ingredient against sunlight exposure [81] with promising applications in dermatology. Mice administered with G-003M before irradiation exhibited elevated survival rates and diminished pulmonary symptoms in contrast to irradiated individuals in the control group. In the context of G-003M dosage and radiation intensity, the observed outcomes delineate a complex cause-and-effect relationship, elucidating heterogeneous survival rates among mice. Notably, an individual intramuscular administration of G-003M, administered 60 min before exposure to a 9 Gy dose, yielded an 89% survival rate in C57BL/6J mice [76]. Further analysis revealed a remarkable 100% survival rate in mice pre-treated with G-003M, juxtaposed with a 66.50% survival rate in those exposed to 11 Gy thoracic gamma radiation (TGR) [77]. This disparity highlights the potential impact of G-003M on survivability under distinct radiation doses. Moreover, this study demonstrated effectiveness in preserving pulmonary vascular integrity while mitigating inflammation and fibrosis, as evidenced by [77]. These multifaceted observations underscore the promising prospects of G-003M as a radioprotective agent, suggesting its nuanced influence on survival rates and pulmonary health in the context of varying radiation exposures. Such insights contribute significantly to the evolving field of radioprotective research, offering valuable considerations for potential clinical applications.

G-002M, comprising PPT, podophyllotoxin-β-D glucoside, and rutin, demonstrated significant protection against radiation-induced hematopoietic suppression and cytogenetic aberrations [55]. Investigations into comprehensive bodily shielding have revealed that a solitary preventative administration of G-002M ensured over 85% survival in mice subjected to a lethal dose of gamma radiation (9 Gy), notably safeguarding the radiosensitive hematopoietic and gastrointestinal organs [55]. Pre-treatment with G-002M preserved bone marrow progenitor cells, maintained myeloid/erythroid ratios, and reduced chromosomal aberrations [82]. Additionally, the formulation stimulated antioxidant responses, minimized DNA damage, and promoted lymphocyte proliferation, indicating its potential to safeguard hematopoietic function post-radiation exposure.

PPT derivatives, exemplified by G-003M and G-002M formulations, offer promising avenues for developing effective radioprotective strategies [55,76]. These compounds exhibit multifaceted mechanisms, including free radical scavenging, DNA protection, and modulation of apoptotic pathways and inflammatory responses. Further research and clinical trials are warranted to validate their safety and efficacy for human application in mitigating the adverse effects of radiation exposure. The summary of the use of PPT in preclinical and clinical studies is shown in Table 3.

#### 3.1.3. Graphical Summary

All the above-mentioned substances, which are derivatives of podophyllotoxin, are summarized in Figure 4. It categorizes them based on their activity: those with only anticancer effects, those with non-anticancer effects, and those exhibiting both types of activity.

#### 3.1.4. Geographical Distribution of Podophyllotoxin Research

Our review of the scientific literature indicates a regional interest in PPT research, particularly in areas where the Podophyllum plant is native. Notably, a significant number of studies from Asia and Latin America have explored the applications of PPT. This pattern suggests that local factors, such as geography, culture, or climate, might influence researchers’ focus on PPT. However, it is critical to conduct further research to confirm these correlations and identify the specific influences on scientific inquiry into PPT and its derivatives. Such studies are vital to understanding the motivations behind PPT research and the role of local conditions in shaping scientific interests globally.

The geographical distribution of research on PPT, as illustrated in the map (Figure 5), further reinforces these regional trends. The map highlights the countries with the highest number of studies, revealing a concentration of research in regions where the Podophyllum plant is native. In particular, China and the United States stand out with 13 studies each, followed closely by India with 10 studies. This concentration may be attributed to the availability of the plant in these regions, as well as the local interest in exploring its applications. The pattern also suggests that environmental, cultural, and climatic factors could play a role in shaping the direction of PPT research in these countries.

While the map provides a general overview of the geographical distribution of PPT research, it is important to note that international collaborations, where authors come from multiple countries, may lead to a broader representation of global research efforts. Thus, the map should be viewed as a general reflection of global research activity rather than an exhaustive account.

Further research is necessary to confirm these correlations and delve deeper into the specific local conditions influencing the scientific interest in PPT. This will be crucial for understanding the broader motivations behind the growing body of PPT research and how local factors shape scientific inquiry in different regions.

### 3.2. Discussion

The findings underscore the diverse effects of PPT and its derivatives, prompting discussions on their therapeutic potential and associated concerns.

A contentious issue arises from the observed reduction in mouse oocyte developmental competence post-PPT exposure, necessitating further research on its reproductive implications and impact on embryonic development. Alterations in organelle distribution and compromised cellular functions following PPT exposure raise safety concerns, particularly regarding its effects on critical cellular processes, such as protein synthesis and vesicle-based transport, during oocyte maturation.

PPT-induced dysfunctions in mitochondria and oxidative stress in embryonic development highlight potential teratogenic effects, demanding comprehensive safety assessments in clinical settings, especially in reproductive contexts.

Despite these concerns, PPT shows promise in treating various non-cancerous conditions, particularly dermatological disorders, and as a radioprotective agent against radiation exposure. These findings align with previous research demonstrating the antiviral and radioprotective properties of podophyllotoxin derivatives. The included studies varied significantly in terms of study design, population, and outcome measures, which limited the ability to perform a quantitative synthesis. Additionally, most studies lacked standardized reporting of effect sizes and confidence intervals, which hinders the direct comparability of results. Despite challenges, PPT remains a valuable therapeutic option, warranting continued investigation for optimized clinical application and patient safety. This review was limited by the exclusion of non-English studies, which may have omitted relevant evidence from other regions. Furthermore, the manual screening and data extraction processes, while thorough, could have introduced subjective biases.

#### Gaps and Future Perspectives

Despite the promising therapeutic potential of PPT and its derivatives, several challenges limit their clinical application. The broader therapeutic potential of these compounds in non-cancerous diseases, such as their applications in pain management and inflammation control, remains underexplored and warrants further investigation. Moving forward, rigorous research is needed to elucidate PPT’s adverse effects and molecular mechanisms, ensuring safe clinical use. The systemic toxicity of PPT underscores the need for further research to establish safe dosage ranges and develop derivatives with improved safety profiles. While the antimitotic and cytotoxic mechanisms of PPT are well-documented, its antiviral, analgesic, anti-inflammatory, and radioprotective effects remain underexplored at the molecular level, limiting the rational design of targeted derivatives. The lack of clinical trials focused on non-cancerous applications, coupled with inconsistent outcome measures and small, homogenous study populations, hampers the development of evidence-based guidelines. Advances in the synthesis of derivatives, such as etoposide, demonstrate the potential for structural modifications to broaden therapeutic applications, yet efforts have largely focused on oncology, leaving non-cancerous uses insufficiently investigated. Furthermore, combination therapies, which show preliminary promise, require systematic exploration to optimize efficacy and minimize toxicity. Policymakers should consider supporting research into these compounds to address unmet medical needs, particularly in resource-limited settings where PPT is readily available. Future studies should aim to establish standardized outcome measures and explore novel derivatives with reduced toxicity profiles. Addressing these gaps through rigorous, inclusive, and interdisciplinary research is essential to fully realize the clinical utility of PPT in non-cancerous diseases.

## 4. Conclusions

Our systematic review explores the complex pharmacological effects of podophyllotoxin (PPT) and its derivatives across various medical domains. PPT demonstrates significant cytotoxic effects through mechanisms like mitochondrial dysfunction, protein metabolism disruption, and apoptosis induction. Its adverse impact on critical embryological processes also highlights the necessity for cautious application in reproductive health.

Additionally, our study reveals the potential of PPT derivatives in treating viral infections, notably their effectiveness against conditions such as genital warts caused by the Human papillomavirus (HPV). The development of novel derivatives, such as cyclolignan SAU-22.107, presents promising avenues in antiviral research, particularly when used in combination therapies with agents like imiquimod.

Furthermore, PPT and its derivatives have shown remarkable analgesic and anti-inflammatory properties, underscoring their potential utility in managing pain and inflammation. Their anti-inflammatory and immunomodulatory effects are particularly noteworthy, suggesting further exploration into autoimmune disorders, viral infections, and dermatological conditions.

The radioprotective properties of derivatives like G-003M and G-002M underscore their capacity to mitigate radiation-induced damage through mechanisms such as free radical scavenging and modulation of apoptotic pathways.

In conclusion, while current research has limitations, our findings emphasize the diverse pharmacological properties and therapeutic potential of PPT and its derivatives. To fully explore their clinical applicability, future research should focus on clinical trials targeting noncancerous applications, such as antiviral treatments and pain management. Additionally, efforts should be directed towards improving the synthesis methods of PPT derivatives to reduce their toxicity and enhance their therapeutic profiles. Continued investigation is essential to confirm their safety, efficacy, and long-term applicability in clinical settings. Nevertheless, our insights significantly contribute to the ongoing discussion on the therapeutic applications of these compounds, setting the stage for future research in this promising field.

## Figures and Tables

**Figure 1 ijms-26-00958-f001:**
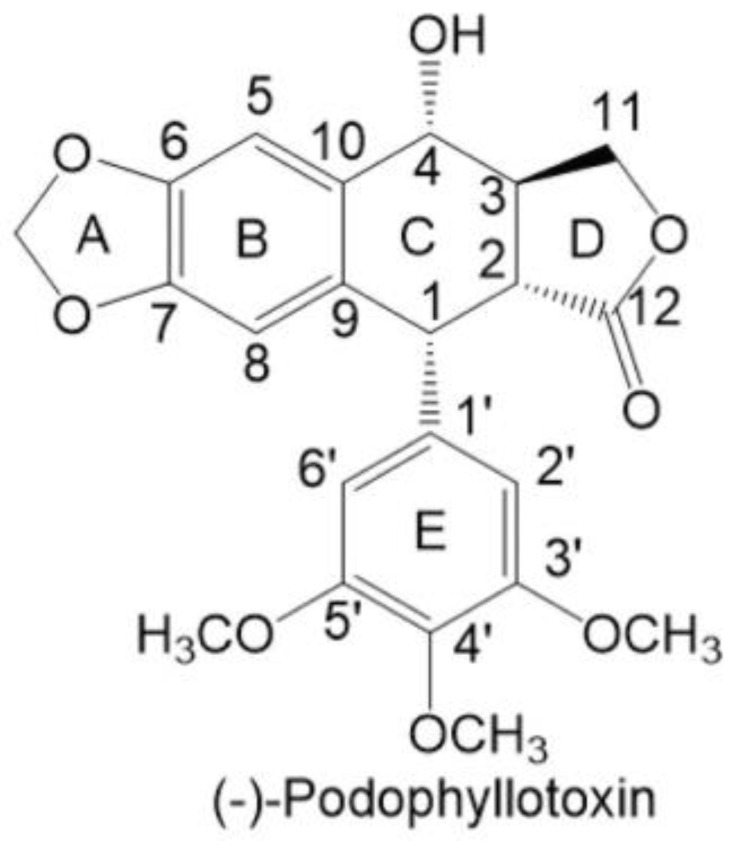
Podophyllotoxin’s structure. Adopted from [8].

**Figure 2 ijms-26-00958-f002:**
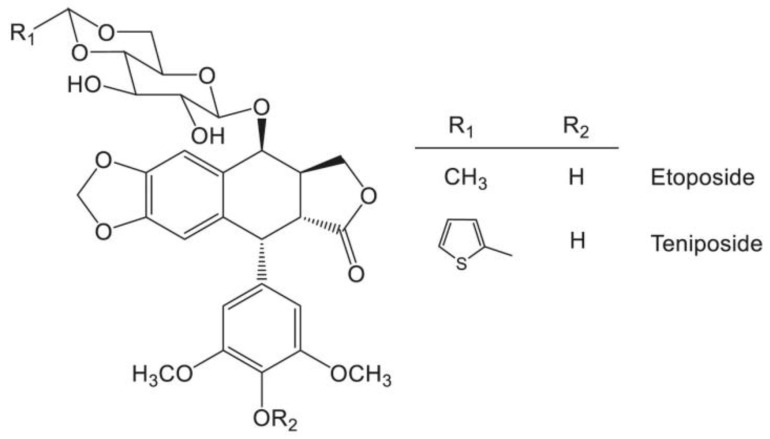
Etoposide and teniposide. Adopted form [17].

**Figure 3 ijms-26-00958-f003:**
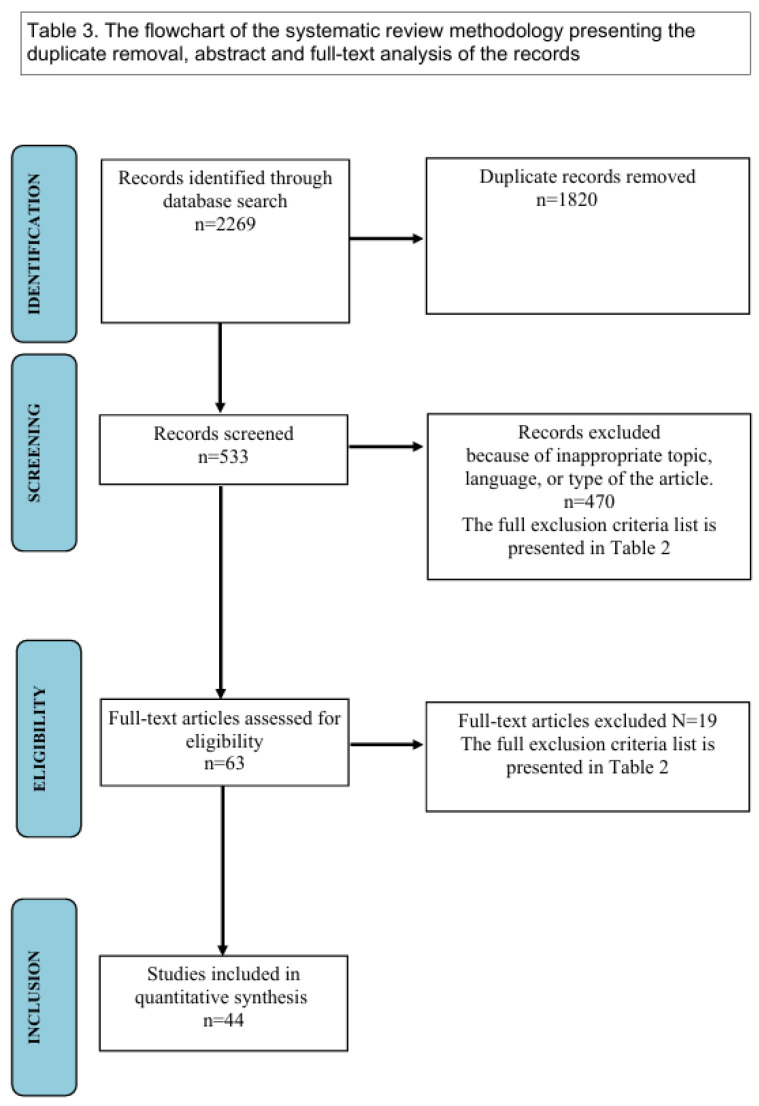
Flow diagram—the selection process of the articles. Adapted from “The PRISMA 2020 statement: an updated guideline for reporting systematic reviews” [48]. Copyright by the British Medical Association.

**Figure 4 ijms-26-00958-f004:**
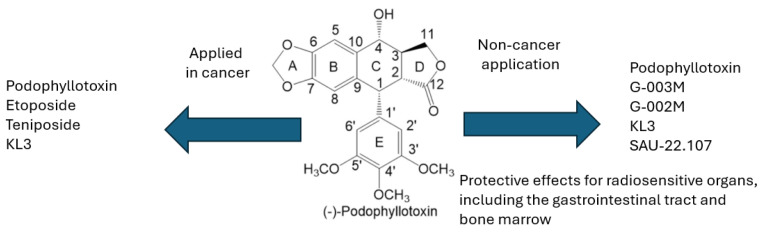
Summary of podophyllotoxin derivatives, categorizing them based on their activity: anticancer effects only, non-anticancer effects, and both anticancer and non-anticancer effects.

**Figure 5 ijms-26-00958-f005:**
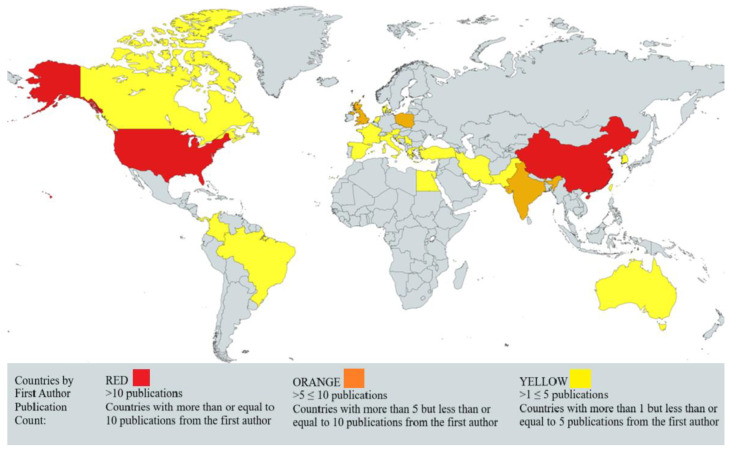
The map illustrates the geographical distribution of Podophyllotoxin (PPT) research based on the affiliation of the first author. The map was created using https://www.mapchart.net/ (accessed on 19 January 2025).

**Table 1 ijms-26-00958-t001:** The combined search strategy for the systematic review.

Database	Number of Records	Search Strategy
PubMed/MEDLINE	408	((podophyllotoxin[Title/Abstract]) OR (ppt[Title/Abstract])) AND ((non-cancerous[Title/Abstract]) OR (antiviral[Title/Abstract]) OR (disease[Title/Abstract])) AND ((“2013/01/01”[Date—Publication]: “3000”[Date—Publication]))
Embase	1104	(‘podophyllotoxin’ OR ‘ppt’) AND (‘non-cancerous’ OR ‘antiviral’ OR ‘disease’) AND [2013–2025]/py AND ‘article’/it
Web of Science	757	TS = (podophyllotoxin OR ppt) AND TS = (non-cancerous OR antiviral OR disease) AND PY = 2013–2025

**Table 2 ijms-26-00958-t002:** The inclusion and exclusion criteria of the articles.

Inclusion Criteria	Exclusion Criteria
Original studies about Podophyllotoxin and its derivatives	Meta-analysis, systematic review, books, guidelines
Language of the article: English	Other unoriginal articles
Describing non-cancerous application and activity	Language other than English
Human, animal, cellular, and molecular studies	Not finished work
Good quality of the research	Articles about only anti-cancerous activity

**Table 3 ijms-26-00958-t003:** Summary of Podophyllotoxin and its derivatives: anticancer applications, non-anticancer applications, special features, and references.

Compound	Anticancer Applications	Non-Anticancer Applications	Special Features	Citation
Compounds Tested in Clinical Settings in Humans				
Podophyllotoxin (PPT)	Treatment of skin cancers (topical)	Treatment of genital warts, psoriasis, molluscum contagiosum, keratoacanthoma; antiviral properties (e.g., HPV, SARS-CoV-2)	Topical use only due to high systemic toxicity; interacts with microtubule spindle formation	[27,28,29,30,31,32,33]
Etoposide	Ovarian, testicular, and lung cancer; leukemia; Hodgkin’s lymphoma; non-Hodgkin’s lymphoma	Not currently used in non-cancer treatment.	Systemic application; a first-line treatment for testicular and small-cell lung cancers	[40,41,42,43,44,45]
Teniposide	Hodgkin’s lymphoma, bladder cancer, acute lymphocytic leukemia, immature neuroblastoma	Not currently used in non-cancer treatment.	Systemic application, similar to etoposide but distinguished by its thienyl group	[46]
Compounds tested in preclinical settings in animals				
G-003M (PPT + rutin)	Not currently used in cancer treatment.	Radioprotection against gamma radiation-induced lung and tissue damage; reduces oxidative stress; enhances survival rates in preclinical models	Administered prophylactically; strong antioxidant properties; preserves pulmonary vascular integrity during radiation exposure	[77,78,79,80,81]
G-002M (PPT, rutin, derivatives)	Not currently used in cancer treatment.	Radioprotection against hematopoietic suppression and chromosomal aberrations; protects bone marrow and reduces DNA damage post-radiation	Single-dose preventative administration; protective effects for radiosensitive organs, including the gastrointestinal tract and bone marrow	[55,76,77,78,79,80,81,82]
Compounds tested in preclinical settings on cells				
KL3	Used in pre-clinical study as anticancer agent for Hela, MDA-MB, MCF7, PC3, DU-145, CFPAC cell lines	Reduces cytotoxicity in keratinocyte models compared to PPT; potential for less toxic therapeutic use	No necrotic effects; activates caspase-9 in keratinocytes	[17,34]
SAU-22.107	Not currently used in cancer treatment.	Decreases replication of Dengue virus; neuroprotective potential against SARS-CoV-2; modulates interferon-regulatory factors to prevent viral replication	Experimental compound; antiviral activity via modulation of viral binding to host cells	[73,74,75]

## Data Availability

The raw data supporting the conclusions of this article will be made available by the authors on request. No template data collection forms, analytic code, or additional materials were used in this review. All extracted data are available upon request.

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
