# Peer review of "The Effects of Podophyllotoxin Derivatives on Noncancerous Diseases: A Systematic Review"

_ijms, 2025, doi:10.3390/ijms26030958_

Round 1
Reviewer 1 Report
Comments and Suggestions for Authors
General Comments:
This systematic study examines the therapeutic potential of podophyllotoxin (PPT) derivatives in noncancerous disorders, a subject often eclipsed by their anticancer uses. The authors conducted a comprehensive literature review, examined pertinent studies, and emphasized both potential uses and constraints. The work possesses merit but would improve with enhanced clarity, organization, and analytical rigor.
Major Comments:
Methodology:
The search technique and inclusion/exclusion criteria are clearly articulated; nevertheless, the PRISMA flow diagram ought to include more comprehensive justifications for the exclusion of research.
The manuscript indicates that two reviewers independently evaluated the articles; nevertheless, it fails to specify the method of measuring inter-rater reliability or the process for resolving discrepancies. Incorporating these facts would enhance methodological rigor.
Results and Discussion:
Although the results are thorough, certain portions require elucidation:
In Section 3.1.3 (Mitotic spindle inhibition), it would be beneficial to illustrate the implications of these discoveries for noncancerous therapeutic applications.
In Section 3.1.5 (Antiviral Properties), molecular findings, such as interactions with SARS-CoV-2 proteins, should be more explicitly linked to possible therapeutic applications.
The discussion could benefit from a more critical analysis of the limitations of the included studies, such as study design biases or small sample sizes.
Figures and Tables:
Figures 1 and 2 are informative but could be enhanced with more detailed captions, explaining their relevance to the manuscript.
Table 3: Table 3 is a good summary of the therapeutic potential of PPT derivatives but should explicitly differentiate between preclinical and clinical findings.
While the conclusion summarizes the findings well, it could be expanded to include specific recommendations for future research directions, such as the need for clinical trials on noncancerous applications or improved derivative synthesis methods to reduce toxicity.
Minor Comments:
Language and Style:
The manuscript is generally well-written but contains occasional grammatical errors and awkward phrasing (e.g., "Podophyllotoxin could be effective toward cancer cells"). Careful proofreading is recommended.
Consistency:
The manuscript inconsistently refers to podophyllotoxin as "PPT" and "PTT." Ensuring uniform terminology throughout would improve readability.
References:
While the reference list is extensive, it could benefit from clearer organization, particularly distinguishing between primary studies and reviews.
Recommendation:
Minor Revision
The manuscript has significant merits and provides valuable insights into the therapeutic applications of PPT derivatives in noncancerous diseases. Addressing the points mentioned above will further strengthen its impact and clarity.
Author Response
Reviewer 1
Comment:
"The PRISMA flow diagram ought to include more comprehensive justifications for the exclusion of research."
Response:
We have expanded the PRISMA flow diagram to include detailed justifications for the exclusion of studies, as per your suggestion. This update is reflected in the PRISMA flow diagram in the Methods section (line 221).
Comment:
"The manuscript indicates that two reviewers independently evaluated the articles; nevertheless, it fails to specify the method of measuring inter-rater reliability or the process for resolving discrepancies. Incorporating these facts would enhance methodological rigor."
Response:
We have clarified the process used to measure inter-rater reliability and described how discrepancies between reviewers were resolved. These details have been added to the Methods section (lines 179-186).
Comment:
"In Section 3.1.3 (Mitotic spindle inhibition), it would be beneficial to illustrate the implications of these discoveries for noncancerous therapeutic applications."
Response:
We have expanded Section 3.1.3 to discuss the potential implications of spindle inhibition for noncancerous therapeutic applications. This addition can be found in lines [329-341].
Comment:
"In Section 3.1.5 (Antiviral Properties), molecular findings, such as interactions with SARS-CoV-2 proteins, should be more explicitly linked to possible therapeutic applications."
Response:
We have revised Section 3.1.5 to explicitly link molecular interactions with SARS-CoV-2 proteins to potential antiviral applications. The updated text is located in lines [393-402].
Comment:
"The discussion could benefit from a more critical analysis of the limitations of the included studies, such as study design biases or small sample sizes."
Response:
We have added a critical analysis of the limitations of the included studies in the Discussion section. This includes a discussion on study design biases and sample size limitations (lines 539 - 541).
Comment:
"Figures 1 and 2 are informative but could be enhanced with more detailed captions, explaining their relevance to the manuscript."
Response:
The captions for Figures 1 and 2 have been expanded to provide more context and explain their relevance. The mentioned figures serve to provide an overview of the chemical characteristics of podophyllotoxin and its similarity to etoposide and teniposide. We have added additional descriptions in the text to emphasize the similarities and differences in their structures and properties. These updates are in lines [143 - 146].
Comment:
"Table 3 is a good summary of the therapeutic potential of PPT derivatives but should explicitly differentiate between preclinical and clinical findings."
Response:
We have revised Table 3 to explicitly differentiate between preclinical and clinical findings, as requested. This revision is reflected in Table 3.
Comment:
"While the conclusion summarizes the findings well, it could be expanded to include specific recommendations for future research directions, such as the need for clinical trials on noncancerous applications or improved derivative synthesis methods to reduce toxicity."
Response:
The Conclusion has been expanded to include specific recommendations for future research directions, focusing on clinical trials and derivative synthesis methods to reduce toxicity. These updates are in lines [588 - 595].
Comment:
The manuscript is generally well-written but contains occasional grammatical errors and awkward phrasing (e.g., "Podophyllotoxin could be effective toward cancer cells"). Careful proofreading is recommended.
Response:
We have thoroughly reviewed the entire manuscript for language mistakes and made corrections to grammar and awkward phrasing.
Comment:
The manuscript inconsistently refers to podophyllotoxin as "PPT" and "PTT." Ensuring uniform terminology throughout would improve readability.
Response:
Thank you for that comment, we have changed each PTT for PPT.
Comment:
While the reference list is extensive, it could benefit from clearer organization, particularly distinguishing between primary studies and reviews.
Response:
All references included in the main analysis are original studies, as this was part of our inclusion criteria. Review articles were only used to provide context in the Introduction section. This ensures that our conclusions are based solely on primary research findings.
Reviewer 2 Report
Comments and Suggestions for Authors
After a thorough review of the manuscript entitled “The effects of podophyllotoxin derivatives on noncancerous diseases – Systematic Review”, I highlighted some points that should be taken into consideration by the authors to improve the study.
1. ABSTRACT: The results presented in the abstract are very limited. I suggest including more information about the main findings of this study. Additionally, a brief conclusion should also be included.
2. METHODOLOGY: I suggest expanding the searches for articles to other databases such as Scopus, ScienceDirect, and SpringerLink. Some important articles may not have been included in this systematic review.
3. This manuscript needs to be updated and the search should cover articles published up to the current month (January 2025).
4. RESULTS: No figures or tables were included in the results section of this study. As this is a review article, the inclusion of these elements in the text is essential to improve the presentation of results and make the text more didactic.
5. In section “3.1.8. Geography”, I suggest including a map indicating the geographic regions that demonstrate interest in Podophyllotoxin research.
6. DISCUSSION: The discussion of this study is very brief and should include more information on the central topic of this article. Authors should draw on previously published articles to expand and improve the discussion.
7. Authors should include a section entitled “Gaps and future perspectives” and report the main challenges encountered for the clinical use of podophyllotoxin.
Author Response
Reviewer 2
Comment:
"The results presented in the abstract are very limited. I suggest including more information about the main findings of this study. Additionally, a brief conclusion should also be included."
Response:
We have revised the Abstract to include more detailed results and a brief conclusion. The updated Abstract can be found in lines [25-26, 28-29, 32-37, 41-44].
Comment:
"I suggest expanding the searches for articles to other databases such as Scopus, ScienceDirect, and SpringerLink. Some important articles may not have been included in this systematic review."
Response:
Thank you for this suggestion. In our review, we searched three major databases (PubMed/MEDLINE, Embase, and Web of Science), which are commonly used and widely accepted as comprehensive sources in systematic reviews. Many systematic reviews rely on searches of these three databases to ensure robust and thorough coverage of the literature. Adding additional databases, such as Scopus, ScienceDirect, or SpringerLink, would require significant time and effort, potentially exceeding the revision deadline. We believe that the current database selection provides sufficient coverage and is unlikely to affect the primary conclusions of our study.
Comment:
"This manuscript needs to be updated and the search should cover articles published up to the current month (January 2025)."
Response:
We have updated our search to include articles published up to January 2025. Methods (number of analyzed studies etc.) was updated – Table 1 (line 169). The 2 newly identified studies (lines: 731, 743) have been incorporated into the results and are reflected in the updated flowchart (line 220), as well as in the corresponding tables summarizing the findings – Table 3 (line 618-619). These updates ensure the manuscript includes the most current and relevant literature.
Comment:
"No figures or tables were included in the results section of this study. As this is a review article, the inclusion of these elements in the text is essential to improve the presentation of results and make the text more didactic."
Response:
We have added new figures 4 and 5 to the Results section to improve the presentation of our findings. These are located in lines [487 and 516].
Comment:
"In section '3.1.8. Geography,' I suggest including a map indicating the geographic regions that demonstrate interest in Podophyllotoxin research."
Response:
We have added a map as figure 5 to Section 3.1.4 to illustrate the geographic regions of interest in Podophyllotoxin research. This is reflected in lines [490-515].
Comment:
"The discussion of this study is very brief and should include more information on the central topic of this article. Authors should draw on previously published articles to expand and improve the discussion."
Response:
The Discussion section has been expanded to include additional analysis and references to previously published articles. This can be found in lines [532 - 537].
Comment:
"Authors should include a section entitled 'Gaps and future perspectives' and report the main challenges encountered for the clinical use of podophyllotoxin."
Response:
We have added a new section titled "Gaps and Future Perspectives" (3.2.1) to address the main challenges for the clinical use of podophyllotoxin. This section is located in lines [542].
Reviewer 3 Report
Comments and Suggestions for Authors
The author's manuscript provides a comprehensive perspective on the use of podophyllotoxin. The author's manuscript has good logic and expression, and clearly and accurately describes its views. On this basis, the author needs to supplement and revise some parts of the manuscript that are not clear and accurate enough to make the manuscript close to perfection,
1. Many references are marked after the period, which may cause doubts. Please adjust the format.
2.Page 3, lines 88-89, "Therefore, despite Podophyllotoxin could be effective toward cancer cells [36] its antineoplastic use is limited to condyloma acuminata topical treatment. [30]". For researchers who are not specialized in the field of condyloma acuminata treatment, they may not be clear about the relationship between the antineoplastic effect and the treatment of condyloma acuminata. Although they are detailed later in the manuscript, it is recommended that the author make a brief explanation here.
3. The author could describe more about podophyllotoxin's treatment of noncancerous diseases in the introduction, such as the reasons for studying this aspect. Currently, the introduction almost exclusively talks about its antineoplastic effects.
4. In section 3.1.1. Toxicity of PTT to non-cancerous cells, the author added the content of derivative KL3 in the explanation of the PPT, but its location makes the logical structure of the article slightly flawed. It is recommended that the author adjust it.
5. In section 3.1. Results, the authors demonstrated the toxic and therapeutic effects of podophyllotoxin and its derivatives on non-cancer cells. These effects are of reference value, but the authors are advised to adjust the directory structure and distinguish between toxic and therapeutic effects to make the article more structured.
Author Response
Reviewer 3
Comment:
"Many references are marked after the period, which may cause doubts. Please adjust the format."
Response:
All references have been reformatted to ensure consistency and correct placement relative to punctuation. These changes are reflected throughout the manuscript.
Comment:
"Page 3, lines 88-89: For researchers who are not specialized in the field of condyloma acuminata treatment, they may not be clear about the relationship between the antineoplastic effect and the treatment of condyloma acuminata. Although they are detailed later in the manuscript, it is recommended that the author make a brief explanation here."
Response:
We have added a brief explanation in lines [87-91] to clarify this relationship for non-specialist readers.
Comment:
"The author could describe more about podophyllotoxin's treatment of noncancerous diseases in the introduction, such as the reasons for studying this aspect."
Response:
The Introduction has been revised to include more details about the rationale for studying podophyllotoxin's treatment of noncancerous diseases (lines [56-61, 87-91, 117-120]).
Comment:
"In section 3.1.1. Toxicity of PTT to non-cancerous cells, the author added the content of derivative KL3 in the explanation of the PPT, but its location makes the logical structure of the article slightly flawed. It is recommended that the author adjust it."
Response:
We have adjusted the placement of the KL3 discussion and added 3.1.1.2 Section to improve logical flow. This can be found in lines [273-282].
Comment:
"In section 3.1. Results, the authors demonstrated the toxic and therapeutic effects of podophyllotoxin and its derivatives on non-cancer cells. These effects are of reference value, but the authors are advised to adjust the directory structure and distinguish between toxic and therapeutic effects to make the article more structured."
Response:
The Results section has been restructured to clearly distinguish between toxic and therapeutic effects. We added section 3.1.1. Toxic effects of PPT and its derivatives and 3.1.2. Therapeutic effects of PPT and its derivatives. This revision is reflected in lines [224 and 360].
Round 2
Reviewer 2 Report
Comments and Suggestions for Authors
The authors responded to my comments and necessary changes were made to the manuscript.